# F1 Male Sterility in Cattle-Yak Examined through Changes in Testis Tissue and Transcriptome Profiles

**DOI:** 10.3390/ani12192711

**Published:** 2022-10-09

**Authors:** Mengli Cao, Xingdong Wang, Shaoke Guo, Yandong Kang, Jie Pei, Xian Guo

**Affiliations:** 1Key Laboratory of Yak Breeding Engineering of Gansu Province, Lanzhou Institute of Husbandry and Pharmaceutical Sciences, Chinese Academy of Agricultural Sciences, Lanzhou 730050, China; 2Key Laboratory of Animal Genetics and Breeding on Tibetan Plateau, Ministry of Agriculture and Rural Affairs, Lanzhou 730050, China

**Keywords:** yak, cattle-yak, testicular tissue, RNA-seq

## Abstract

**Simple Summary:**

Cattle-yak, a crossbreed of cattle and yak, has evident heterosis but F1 male cattle-yak is unable to generate sperm and is sterile, which limits the fixation of heterosis. This study analyzed the differences in testicular tissue development between four-year-old yak and cattle-yak from the perspective of histomorphological changes and sequenced the testicular tissue of the two using RNA-seq technology, examining the differential gene expression related to spermatogenesis and apoptosis. These findings offer a theoretical explanation for the sterility in F1 male cattle-yak that can help yak hybridization.

**Abstract:**

Male-derived sterility in cattle-yaks, a hybrid deriving from yak and cattle, is a challenging problem. This study compared and analyzed the histomorphological differences in testis between sexually mature yak and cattle-yak, and examined the transcriptome differences employing RNA-seq. The study found that yak seminiferous tubules contained spermatogenic cells at all levels, while cattle-yak seminiferous tubules had reduced spermatogonia (SPG) and primary spermatocyte (Pri-SPC), fewer secondary spermatocytes (Sec-SPC), an absence of round spermatids (R-ST) and sperms (S), and possessed large vacuoles. All of these conditions could have significantly reduced the volume and weight of cattle-yak testis compared to that of yak. RNA-seq analysis identified 8473 differentially expressed genes (DEGs; 3580 upregulated and 4893 downregulated). GO (Gene Ontology) and KEGG (Kyoto Encyclopedia of Genes and Genomes) enrichment evaluations for DEGs found their relation mostly to spermatogenesis and apoptosis. Among the DEGs, spermatogonia stem cell (SSCs) marker genes (*Gfra1*, *CD9*, *SOHLH1*, *SALL4*, *ID4*, and *FOXO1*) and genes involved in apoptosis (*Fas*, *caspase3*, *caspase6*, *caspase7*, *caspase8*, *CTSK*, *CTSB* and *CTSC*) were significantly upregulated, while differentiation spermatogenic cell marker genes (*Ccna1*, *PIWIL1*, *TNP1*, and *TXNDC2*) and meiosis-related genes (*TEX14*, *TEX15*, *MEIOB*, *STAG3* and *M1AP*) were significantly downregulated in cattle-yak. Furthermore, the alternative splicing events in cattle-yak were substantially decreased than in yak, suggesting that the lack of protein subtypes could be another reason for spermatogenic arrest in cattle-yak testis.

## 1. Introduction

The yak (*Bos grunniens*), a bovid species of the Qinghai-Tibet Plateau (QTP) and adjacent regions [1], lives at an altitude of 3000–5000 m and an annual average temperature of −5 to −1 °C. It provides local herders with meat, wool, milk, labor, fuel, and other everyday essentials. Cattle-yak, a cross breed between cattle and yak, has the same adaptability as yak and can also adapt to harsh environments (high altitude, low temperatures, and hypoxia) [2]. In fact, cattle-yak has made a greater contribution to the progression of animal husbandry in the QTP region. It is superior to yak in milk yield, meat products, and other economic traits [3]. However, the F1 cattle-yak males are sterile, while the females are fertile. F1 male cattle-yak have regular reproductive organ morphology and sexual activity as their fertile male counterparts, though lack mature sperm (S) in the semen fluid causing infertility [4]. With the use of grading up, fertility can be restored in F4 generation male cattle-yak [5]; however, it loses heterosis, causing a reduction in production performance and commercial value. Male-derived sterility in cattle-yak obeys the Haldane rule [6], i.e., an individual of a certain sex in the hybrid generation produced by interbreeding between heterogeneous animals cannot survive or is sterile and must be a heterozygous sex. Incompatibilities in interspecific hybrids restrict the gene flow between sexually reproducing organisms, which inhibits the fixation of superior gene combinations in the F1 generation. Therefore, understanding the mechanism of male-derived sterility in cattle-yak has both theoretical and practical significance.

Many attempts have been made to explore the stagnation of spermatogenesis in cattle-yak. Karyotyping of testicular primary spermatocyte (Pri-SPC) showed that cattle-yak has a similar amount of chromosomes (2n = 60) when compared to cattle and yak [7], while the majority of cattle-yak Pri-SPCs comprise the structurally malformed synaptonemal complex of autosomes and XY bivalents are absent [8]. It is speculated that the arrest of meiosis I is the prime cause of infertility in male cattle-yak [9]. Consequently, to understand male-derived sterility in cattle-yak at the molecular level, scholars selected some genes related to meiosis and spermatogenesis to explore variations in their expression levels among yak and cattle-yak. They observed that the testis-specific *Dmrt7* gene expression was significantly decreased in cattle-yak [10]. *DAZL* and *DAZ* involved in spermatogenesis were not expressed and their extent of methylation was substantially elevated than that in cattle and yak [11]. Likewise, the expression levels of the meiosis-related genes *BOULE*, *Rad51*, *DMC1*, *Cdc2*, *Cdc25A*, *bMei1*, and *SYCP3* were significantly reduced within cattle-yak testicular tissue, and methylation levels for *bMei1*/*SYCP3* were significantly higher than in yak [12,13,14,15,16].

Spermatogenesis is regulated by multiple genes and the mechanism of male infertility caused by spermatogenic arrest has not been systematically examined so far. Wu et al. [17] employed RNA-seq to compare the testis gene expression profiles of twelve-month-old cattle-yak and yak before sexual maturity and found 6477 differentially expressed genes (DEGs), which included 2919 upregulated and 3558 downregulated genes. It is speculated that spermatogenic arrest within cattle-yak originates during the differentiation phase concerning spermatogonia stem cell (SSCs) and is exacerbated during spermatogonia (SPG) mitosis and spermatocyte (SPC) meiosis. The sexual maturity age of yak is 2–3 years. Accordingly, here, RNA-seq technology was employed to explore the variations in gene expression profiles between cattle-yak and yak after they reached sexual maturity. Additionally, we analyzed the testicular tissue developmental differences between cattle-yak and yak based on morphological and histological changes. The purpose of the present research was to comprehend the spermatogenic arrest mechanism in cattle yaks in greater detail.

## 2. Materials and Methods

### 2.1. Ethics Statement

The entirety of animal experimental investigations was carried out in line with directives issued by the China Council on Animal Care and the Ministry of Agriculture of China. The Animal Care and Use Committee of the Lanzhou Institute of Husbandry and the Pharmaceutical Sciences Chinese Academy of Agricultural Sciences, Lanzhou, China accepted all animal-handling protocols for this investigation (Permit No: SYXK-2014-0002).

### 2.2. Animals and Sample Collection

Three male Gannan yaks (Y1, Y2, and Y3) were from Linxia County, Linxia Hui Autonomous Prefecture. Three male cattle-yaks (C1, C2, and C3; F1 offspring of Jersey cattle (♂) and Gannan yak (♀)) were from Xiahe County, Gannan Tibetan Autonomous Prefecture. Each animal was 4 years old, and all samples were collected on 10 June 2021.

Yaks were euthanized and the testis tissues were collected. The testicles of cattle-yak were obtained by surgical castration after local disinfection. The wounds were then sutured with surgical needles and cattle-yak was provided penicillin/streptomycin as prophylaxis.

To get rid of the blood stains and impurities, phosphate-buffered-saline was utilized for washing the testicular surface. Observations were recorded regarding the testicular tissue surface, and the data were collected for testicular circumference, testicular length, and testicular weight. Next, the testicular tissue was placed into a clean petri dish and cut longitudinally using a scalpel. The intermediate tissue perpendicular to the long axis was removed and cut into several small tissue blocks. One such block was fixed with testis tissue fixative for slice preparation, and the others were promptly frozen and stored in liquid nitrogen for subsequent RNA-seq analysis. Samples were dispatched to Lanzhou Institute of Husbandry and Pharmaceutical Sciences, Chinese Academy of Agricultural Sciences, Lanzhou, China.

### 2.3. Histological Analysis of Testis Tissue

The excised testicular tissue pieces were placed in the testis tissue fixative solution, and the fixative solution was changed at the 6 h, 12 h, and 24 h, and then soaked in 75% alcohol. After dehydration with an automatic tissue dehydrator and after embedding in paraffin, the paraffin block was cut into 6 μm thick sections with a microtome. The improved HE staining kit^®^ (SolarBio™, Beijing, China) was employed following guidelines provided by the manufacturer. Following staining, neutral gum was used for sealing the sections and a Pannoramic 250^®^ digital section scanner (Drnjier, Jinan, China) was employed for obtaining images. The images were analyzed and cropped using Caseviewer (3DHISTECH Ltd., Budapest, Hungary) software. The statistics of testis histological parameters were performed by randomly selecting 5 non-repetitive fields under the 20× magnification in 6 sections. Results were reported as mean ± standard deviation.

### 2.4. RNA Isolation and Library Preparation

The TRIzol reagent was utilized to extract total RNA. The NanoDrop 2000 Spectrophotometer (Thermo Scientific, Waltham, MA, USA) was employed for assessing the RNA’s quantification and purity, and Agilent 2100^®^ Bioanalyzer (Agilent Technologies™, Santa Clara, CA, USA) probed RNA integrity. Finally, the library was created as per the manufacturer’s manual by employing the TruSeq Stranded mRNA LT Sample Prep^®^ (Illumina™, San Diego, CA, USA). Transcriptome sequencing and analysis were performed by OE Biotechnology Co., Ltd. (Shanghai, China).

### 2.5. RNA Sequencing and DEGsAnalysis

On the Illumina™ HiSeqTM 2500^®^ system, libraries were sequenced, with 150 bp paired-end reads produced. For every sample, approximately 46.87 M raw reads were obtained. Raw reads in FASTQ format were initially processed employing Trimomatic [18], including (1) de-linking, (2) removing low-quality reads, (3) removing low-quality bases from 3′/5′ tails through multiple methods, (4) statistically original sequencing quantity, effective sequencing, Q30 and GC content for detailed assessment. Finally, approximately 46.03 M clean readings were retained per sample. Clean reads were mapped onto genome (version: LU_Bosgru_v3.0 (Lanzhou University, Lanzhou, China)) using HISAT2 [19]. Fragments Per Kilobase of exon Model per million mapped fragments (FPKM) [20] for individual genes was calculated using Cufflinks [21], and read counts for each gene were obtained by HTSeqcount [22]. DESeq (2012) in the R package [23] was employed to perform a differential expression analysis depending upon *q* < 0.05 together with |log2 (foldchange)| > 1. Principal components analysis (PCA) between two sets of samples was performed using the R package vegan. The R package stats was used to draw boxplots and correlation cluster heatmaps for each sample gene. To highlight the patterns of gene expression in various groups and samples, systematic clustering analysis of DEGs was carried out. Gene Ontology (GO) enrichment and Kyoto Encyclopedia of Genomes (KEGG) pathway enrichment evaluations for DEGs were executed in the R package, depending upon hyper-geometric distribution; the filter condition was *p* ≤ 0.05.

After assembling the reads by StringTie (Johns Hopkins University, Baltimore, MD, USA) [24] software, ASProfile [25] was used to analyze the alternative splicing (AS) of differentially regulated transcripts, isoforms, or exons.

### 2.6. RNA Sequencing Validation

Genomic DNA was isolated and reverse transcribed according to the instructions of the Transcriptor First Strand^®^ cDNA synthesis kit (Takara Bio Inc.™, Beijing, China). Quantitative reverse transcription-polymerase chain reaction (RT-qPCR) was employed for detecting the DEGs expression levels in two groups of samples. Each group included three biological replicates and three technical replicates per sample. Reverse transcription primers (Appendix A) were designed using the NCBI website and the expression levels were normalized with glyceraldehyde-3-phosphate dehydrogenase (*GAPDH*) as the reference gene. Furthermore, qRT-PCR reaction volume was 20 µL, containing 10 µL 2× PrecisionPLUS^®^ Master Mix (Primerdesign™, Camberley, UK), 1 µL cDNA (25 ng), 1 µL each of primers (F/R) (300 nmol) and 7 µL nuclease-free water. After undergoing incubation at 95 °C for 3 min, the reactions underwent 39 cycles at 95 °C and 55 °C for 10 and 30 s, respectively. The 2^−∆∆Ct^ technique was utilized for calculating the relative expression.

### 2.7. Statistical Analyses

SAS 9.4^®^ was employed in conducting all statistical assessments; * *p* < 0.05, ** *p* < 0.01 denoted statistically significant variations.

## 3. Results

### 3.1. Histomorphological Characteristics of Yak and Cattle-Yak Testis

At superficial level within testis tissue, the blood vessels in the vascular capsule of the cattle-yak testis were small and sparsely and disorderly distributed (Figure 1a,b). The estimation of testicular morphological indicators showed that the circumference, length, and weight of the testis were significantly smaller in cattle-yak than in yak (Table 1).

Concerning the histology of testis tissue, compared to yak, the seminiferous tubules within cattle-yak testes were sparse, shrunken, having large distances between the tubes, with large vacuoles (Figure 1c,d). From the perspective of germ cell distribution, the seminiferous tubules of yak (Figure 1e) had various germ cells from the basement membrane to the lumen, including SPG, Pri-SPC, Sec-SPC, round R-ST and S in seminiferous tubules. However, seminiferous tubules of cattle-yak had reduced numbers of SPG and Pri-SPC, less Sec-SPC, and absence of R-ST and S (Figure 1f). Compared with yak, cattle-yak seminiferous tubules were smaller in diameter with thinner walls. In addition, SPG and Pri-SPC diameter were smaller than yak (Table 2) and Pri-SPC s were vacuolated showing large vacuoles in the cytoplasm with a lighter color. To sum up, the testicular development of cattle-yak was abnormal.

### 3.2. Sequencing Quality and Alignment

In this study, fresh RNA libraries of cattle-yak (*n* = 3) and yak (*n* = 3) testicular tissues were sequenced through the Illumina™ HiSeqTM 2500^®^ system, generating 150 bp paired-end reads. Overall, 42.17 G of raw reads were collected, with final data-volume/sample distributed between 45.02–46.68 M; the Q30 base was distributed between 92.95–93.33%, and the average GC content was 50.19% (Appendix A). Approximately 94% of the clean-reads were mapped onto reference genome (version: LU_Bosgru_v3.0); the mapped unique reads ranged from 36,951,818 to 38,748,921, with the rate of 82.03 to 83.15%. The ratio of multiple mapped reads was 10.64–12.10% (Table 3).

### 3.3. Protein-Coding Gene Expression Analysis

The expression level of genes was estimated based on the FPKM value (Appendix A). The boxplots of FPKM values of sample genes were consistent in height (Figure 2a,), indicating relatively similar overall gene expression levels among different samples within the same group. Likewise, the number of genes expressed with different expression levels was also consistent in different samples within the same group (Figure 2b). Correlation analysis indicated that the correlation coefficients of different samples were considerably increased within the same group than between the two groups (*p* < 0.01) (Figure 2c). PCA showed that yak clustered together, and cattle-yak clustered together (Figure 2d). In all, sequencing data showed high repeatability and reliability.

### 3.4. Identification of DEGs

DEGs were assessed depending upon *q* < 0.05 together with |log2 (foldchange)| > 1. A total of 8473 DEGs were obtained between the yak and cattle-yak of which 3580 upregulated and 4893 downregulated genes were found in the testis tissues of cattle-yak (Appendix A, Figure 2e). Genes with similar expression patterns are usually functionally related. Hierarchical clustering of DEGs demonstrated that the DEGs of C (cattle-yak) together with Y (yak) groups clustered separately in two groups (Figure 2f). This again demonstrated a considerable difference in the gene expression pattern of yak and cattle-yak testis. Interestingly, the top 30 DEGs with >−9.31 log2-fold change were all downregulated, including *CSNKA2IP* (alpha prime interacting protein), *Rlim* (X-linked gene), *SPATA19* (spermatogenesis associated 19), *TSPAN8* (tetraspanin 8), *SPADH1* (spermadhesin-10), *SPZ1* (spermatogenic leucine zipper 1), *PRSS58* (serine protease 58), *Ubqlnl* (ubiquilin gene 5), and *TNP2* (transition protein 2). All of these genes are related to spermatogenesis, sperm maturation, and sperm motility.

Spermatogenesis is a complex and orderly process involving many regulatory genes. We found that SSCs marker genes, such as *Gfra1* (GDNF family receptor alpha 1), *CD9* (CD9 molecule), *SOHLH1* (Spermatogenesis-and Oogenesis specific Basic Helix-loop-helix-containing Protein 1), *SALL4* (transcription factors Sal-like 4), *ID4* (inhibitor of DNA binding 4), and *FOXO1* (Fork-head box protein O1), were significantly upregulated in cattle-yak testis tissue. On the contrary, the marker genes of differentiated spermatogenic cells, such as *Ccna1* (cyclin A1), *PIWIL1*(piwi-like RNA-mediated gene silencing 1), *TNP1* (transition protein 1), and *TXNDC2* (thioredoxin domain containing 2), were significantly downregulated in cattle-yak testis. Additionally, Some DEGs are related to meiosis, such as *TEX14* (testis expressed 14, meiosis and synapsis associated), *TEX15*, *MEIOB* (meiosis specific with OB-fold), *STAG3* (stromal antigen 3), and *M1AP* (meiosis 1 associated protein), were significantly downregulated in cattle-yak testis tissue. In somatic cells, sertoli cell (SC) marker genes BEX2 (brain expressed X-linked 2, WFDC2(WAP four-disulfide core domain 2), ITGA6 (integrin subunit alpha 6), except WFDC2, were significantly upregulated in cattle-yak testis tissue; Leydig cells (LC) marker genes, such as IGF1 (insulin like growth factor 1), IGFBP3 (insulin-like growth factor binding protein 3), IGFBP5, CYP11A1 (cytochrome P450 family 11 subfamily A member 1),CYP17A1, INHBA (inhibin subunit beta A), ARX (aristaless related home-obox), and TCF21 (transcription factor 21), were significantly upregulated except for IGFBP3; Myoid cell (MC) marker genes ACTA2 (actin alpha 2), MYH11 (myosin heavy chain 11), GLI1 (GLI family zinc finger 1), etc., were significantly upregulated in cattle-yak testis tissue. The apoptosis-related genes, such as *Fas* (Fas cell surface death receptor), *caspase 3*, *caspase 6*, *caspase 7*, *caspase 8*, *CTSB* (cathepsin B), *CTSK*, and *CTSC*, were significantly upregulated in cattle-yak testis tissue.

### 3.5. GO and KEGG Pathway Enrichment Evaluations for DEGs

To further understand variations in testicular development between yak and cattle-yak, we performed GO and KEGG enrichment assessments for DEGs. GO enrichment analysis describes the distribution of genes in biological process (BP), cellular component (CC), and molecular function (MF). The upregulated DEGs in cattle-yak showed significant enrichment for 661 GO items, including BP: regulation of cell migration, semaphorin-plexin signaling pathway, and positive regulation of NF-kappaB transcription factor activity; CC: collagen-that contains extracellular matrix, extracellular matrix, and cell surface; MF: protein homodimerization activity, GTPase activity, extracellular matrix structural constituent, etc. (Appendix A, Figure 3a). The downregulated DEGs in cattle-yak were significantly enriched in 300 GO entries, including BP: spermatogenesis, cilium assembly, and flagellated sperm motility; CC: cilium, axoneme, and motile cilium; MF: dynein light chain binding, ATP-dependent microtubule motor activity, minus-end-directed, dynein intermediate chain binding, etc. (Appendix A, Figure 3b). Obviously, the downregulated genes in the testis of cattle-yak were primarily enriched within GO entries associated to spermatogenesis, cilia assembly, and sperm motility.

The results of KEGG enrichment evaluations demonstrated upregulated DEGs in cattle-yak testis were significantly enriched for 122 pathways, mainly involving focal adhesion, the MAPK signaling pathway, ECM–receptor interaction, the PI3K-Akt signaling pathway, apoptosis, Wnt and FOXO signaling pathway, etc. (Appendix A, Figure 3c). The downregulated DEGs in cattle-yak were significantly enriched in 35 pathways, mainly involving Oocyte meiosis, Purine metabolism, Glycerophospholipid/Thiamine metabolism, etc. (Appendix A, Figure 3d). In conclusion, the upregulated genes in the testis of the cattle-yak were primarily augmented in the pathways associated with cell migration, proliferation, differentiation, apoptosis and mediating spermatogenesis, and the genes that were downregulated were found to be primarily enriched in the metabolism-related pathways.

### 3.6. RT-qPCR Validation

To validate the RNA-seq data, we selected six DEGs (YIPF1, ATG10, SUN2, LIPG, SDK1, and SAXO1) for RT-qPCR. The outcomes revealed that their expression patterns were in line with the RNA-seq data (Figure 4a).

### 3.7. Alternative Splicing Analysis

Alternative splicing (AS) enables a gene to produce multiple mRNA transcripts, thereby greatly enriching the diversity of encoded proteins. Based on the annotation data of gene structure, we analyzed the AS events in yak and cattle-yak, involving 12 AS types: XSKIP, XMSKIP, XMIR, XIR, XAE, TTS, TSS, SKIP, MSKIP, MIR, IR, and AE (Figure 4b). The TSS type AS events were the most abundant, followed by the TTS type. Overall, cattle-yak (119,207 ± 3304) had fewer AS events than yak (147,777 ± 3711) (*p* < 0.01).

## 4. Discussion

Compared with other technologies, RNA-seq provides better resolution, sensitivity, and a dynamic detection range of >6 orders of magnitude, and it is now considered as an extremely essential tool for analyzing DEGs and mRNA variable splicing events at the transcriptome level [26]. Cattle-yak has excellent heterosis in terms of growth rate, disease resistance, meat quality, etc. However, the fixation of heterosis were limited by the sterility of F1 male cattle-yak due to inability to generate sperm. Therefore, this has become of great interest to the scientific community. This study compared the histomorphological parameters of the testis tissue between four-year-old cattle-yak and yak and analyzed the difference at the transcriptome level by RNA-seq. We found significant information about the stagnation of spermatogenesis in cattle-yak.

We examined the spermatogenic cell development in the testis of sexually matured yak and cattle-yak. Yak testicular tissue had spermatogenic cells at all levels, while cattle-yak seminiferous tubules had a reduced number of SPGs and Pri-SPCs, fewer Sec-SPCs, absence of R-ST and S, and contained large vacuoles. We speculate that the reduction of spermatogenic cells and the existence of large vacuoles may be the reasons for the significant decrease in the volume and weight of cattle-yak testis. Wang et al. [27] observed the testicular tissue of yak before and after sexual maturity and found that meiotic cells appeared at the age of 8 months when some spermatogenic tubules contained R-ST; spermatogenic tubules in twenty-four-month-old yak testis contained all types of spermatogenic cells, SPGs, Pri-SPCs, Sec-SPCs, R-ST, and S. Cai et al. [28] and Wu et al. [17] compared the testis tissue of twelve-month-old yak and cattle-yak and found that yak testis contained spermatogenic cells of all levels, while cattle-yak testis mainly had SPG and large vacuoles between the seminiferous tubules. Combining the outcomes of this research with earlier studies, it seems that the death of spermatogenic cells occurs in cattle-yak at the age of 12 months and can even begin early at the age of 8 months.

In this study, statistical analysis of DEGs found that most of the top 30 DEGs were related to spermatogenesis, sperm maturation, and sperm motility. *SPATA19*, a small protein of 154 amino acids, is expressed in haploid sperm cells [29]. Rat *SPATA19* locates in the mitochondria in the center of mature sperm [30] and is speculated to participate in mitochondrial organization and/or function. Mi et al. [31] constructed a germ cell-unique *SPATA19* knockout murine-model and observed male mice to be infertile as the sperm mitochondria did not function properly. *SPZ1* is a bHLH-Zip transcription factor, Hsu et al. [32] found that *SPZ1* is specifically expressed within testis/epididymis during spermatogenesis in mice and affects spermatogenesis through controlling cellular proliferation/differentiation through bonding with exact DNA sequences as alternative bHLH-Zip molecular players. Downregulation of a series of genes, such as *SPATA19* and *SPZ1*, may hinder spermatogenesis in cattle-yak.

Spermatogenesis is a complex process wherein diploid SSCs undergo differentiation to produce haploid spermatozoa through mitotic, meiotic, and spermatogenic processes that are tightly controlled within transcriptional, post-transcriptional, and translational levels [33]. *Gfra1*, *CD9*, *SOHLH1*, *SALL4*, *ID4* and *FOXO1* were marker genes of SSCs [34,35,36], all upregulated within cattle-yak testes. However, SPC marker genes *Ccna1* [37] and *PIWIL1* [38] were downregulated, and ST marker genes *TNP1* [39] and *TXNDC2* [40] were almost not expressed, which indicates that undifferentiated spermatogonia accumulate in the testis of the cattle-yak, whereas differentiated spermatogenic cells decrease. The completion of meiosis is a crucial step in spermatogenesis and requires a large number of protein structures. Loss of *TEX15* leads to meiotic arrest [41,42], since this protein is required for successful formation of the synaptonemal complex, repair of double strand break, and meiotic recombination. Greenbaum et al. found that *TEX14*−/− mutant male mice do not form intercellular bridges [43], and Sironen et al. found that *TEX14* is also essential for porcine spermatogenesis [44]. *STAG3* is a part of the axial/lateral element of the synaptic complex and has a function in sister chromatid formation in meiosis I [45]. *MEIOB* has single-stranded DNA binding and exonuclease activity and is essential for meiotic recombination [46]. Most SC in *M1AP*-deficient mice were arrested before meiotic I metaphase [47]. The downregulation of this series of genes makes the meiosis of cattle-yak PSC unable to proceed normally. We speculate that the block of cattle-yak spermatogenesis starts from the PSG produces Pri-SPC and is aggravated during the meiotic of Sec-SPC to form R-ST.

Testosterone (T) secreted by Leydig cells plays a key role in the development of the male reproductive system and the maintenance of normal spermatogenesis. We found that the expressions of CYP11A1 and CYP17A1 were significantly upregulated in the testis tissue of cattle-yak. Sato et al. [48] found that the expression of cattle-yak 3β-hydroxysteroid dehydrogenase (3β-HSD) was increased and the expression of androgen receptor (AR) was decreased. T is cholesterol under the regulation of STAR (steroidogenic acute regulatory protein), its side chain is cleaved by CYP11A1 to form pregnenolone, and then 3β-HSD is converted into progesterone, which is finally catalyzed by CYP17A1 in the endoplasmic reticulum, and then combined with AR binding comes into play [49]. Therefore, we speculate that cattle-yak LC secrete more T, but the T concentration in seminiferous tubules is low due to the decrease of AR expression. Narula et al. [50] found that excessive secretion of T can prompt SC to secrete inhibin A, which selectively inhibits the synthesis and secretion of FSH in the anterior pituitary gland. So, the imbalance of hormone regulation in seminiferous tubules is an important cause of spermatogenesis block. Apoptosis is a major regulating factor of germ cell fate in the mammalian testis, and its dysregulation can disturb spermatogenesis [51]. Caspases regulate apoptosis via mitochondrial and death receptor pathways. Caspase 8 is directed to the death receptor complex in the death receptor pathway, and its involvement in the regulation of apoptosis in testicular germ cells through ligand binding (e.g., *TNFα* (tumor necrosis factor alpha), *FasL* (Fas ligand)) has been identified [52]. It is currently recognized that *Fas* are mainly expressed on reproductive cells and its ligand *FasL* is expressed in SC, and the Fas system mediates the apoptosis regulation of SC on spermatogenic cells. Cathepsins, also belonging to the cysteine protease family, can be released from the lysosome to the cytoplasm in response to a certain signal, which can activate the activation of caspases or the release of mitochondrial pro-apoptotic factors, thereby inducing apoptosis [53]. In this study, the expressions of *Fas*, *caspase 3*, *caspase 6*, *caspase 7*, *caspase 8*, *CTSB*, *CTSK*, and *CTSC* were significantly upregulated in the testis tissue of cattle-yak, which may lead to increased apoptosis of spermatogenic cells, decreased weight of cattle-yak testis, and poor development.

GO enrichment analyses of DEGs showed that significantly downregulated DEGs in cattle-yak were mainly enriched for spermatogenesis, cilia assembly, together with sperm motility. KEGG results demonstrated up-regulated DEGs in cattle-yak to be mainly enriched within pathways related to cell migration, proliferation, differentiation, apoptosis and mediating spermatogenesis. Wnt signaling is critical for spermatogenesis. Kerr et al. [54] increased and decreased Wnt signaling levels in mice by a certain induction and found all mice exhibited disruption of spermatogenesis, including germ cell apoptotic activity, together with rapid-loss/dysregulated blood–testis barrier proteomic distribution/morphologies. PI3K-Akt can regulate the expression of mTOR signaling and its downstream target protein p70S6K/4EBP1 through different molecular pathways [55], while mTOR-p70S6K/4EBP1 signaling molecule takes part in regulating the testicular cell proliferation, differentiation and spermatogenesis [56]. The dynamics of adherens junctions and tight junctions are regulated by the MAPK signaling pathway, which is also crucial for meiosis and Sertoli cell proliferation [57]. ECM-receptor interaction helps germ cells to withstand tension, and it hinders their migration [58]. Numerous physiological processes, such as cellular differentiation, apoptotic activity, cellular proliferation, DNA injuries, and repairing activity, as well as its role as an oxidative stress regulator, are all carried out by FOXO [59]. Designated genes of the FOXO family are thought to be extracellular ligands, such as *FasL*, *TRAIL* (TNF-associated apoptosis-inducing ligand), and *TRADD* (TNF-receptor type 1 associated death domain), together with intra-cellular apoptosis-linked constituents including *Bim* (bcl-2 interaction mediator of cell death), pro-apoptosis *bcl-2* family members, and *Bcl-6* [60,61]. FOXO protein translocates to the nucleus and upregulates several target genes encouraging cell cycle arrest, stress resistance, and apoptosis [62]. Thus, the FOXO signaling pathway may promote apoptosis of testicular germ cells.

Alternative RNA splicing involves the joining of exons of RNA transcripts or pre-mRNAs to produce different mRNA encoding different protein subtypes. These protein isoforms with different structures, functions, localization, and other properties play an important role in health and diseases [63]. A study suggests that up to 94% of human genes are subjected to AS [64]. The same has been detected in the fly, nematodes, and other vertebrates [65,66]. AS has been proposed as the main driving force for the evolution of phenotypic complexity in creatures. Testicular tissue is the richest in terms of AS events. Here, we found that the incidence of AS events was considerably less in cattle-yak than in yak, potentially causing the spermatogenic arrest in the former.

Although there are many studies on male sterility in cattle-yak, most of them are single omics or a combination of two omics. In order to systematically and comprehensively analyze the molecular mechanism of male sterility in cattle-yak, a systematic and comprehensive study of male sterility in cattle-yak should be carried out in combination with all omics, including epigenomics, proteomics, and metabolomics. Epigenetic regulation not only regulates the development of the male reproductive system but also controls the early stages of embryonic development and the process of gametogenesis, impaired epigenetic dysregulation disrupts human spermatogenesis, leading to male infertility and related spermatogenesis disorders [67]. Phakdeedindan’s research showed that during the maturation of yak, the regulation of AcK9 gene and the constant level of DNA methylation are both necessary conditions for spermatogenesis, and the inappropriate expression of AcK9 and DNA methylation may be the main factors affecting the normal development of cattle-yak germ cells factor [68]. In the next step, we should integrate all the data on male sterility in cattle-yak and explore the molecular mechanism of male infertility in cattle-yak as much as possible.

## 5. Conclusions

The present research compared and analyzed the differences in testicular tissue development between sexually matured cattle-yak and yak. We believe that reduction in spermatogenic cells together with the existence of large vacuoles within cattle-yak testes are prime reasons for the reduction in weight and volume of cattle-yak testis. RNA-seq was used for analyzing the differences between the transcriptome profiles of yak and cattle-yak testis tissue. The SSCs marker genes and pro-apoptotic-related genes were upregulated, while differentiated spermatogenic cell marker genes were downregulated. We speculate that spermatogenic stagnation in cattle-yak begins as the SPG produces Pri-SPC and is aggravated during the meiotic of Sec-SPC to form R-ST. Meanwhile, the lack of AS events can be another reason for the stagnation of spermatogenesis in cattle-yak.

## Figures and Tables

**Figure 1 animals-12-02711-f001:**
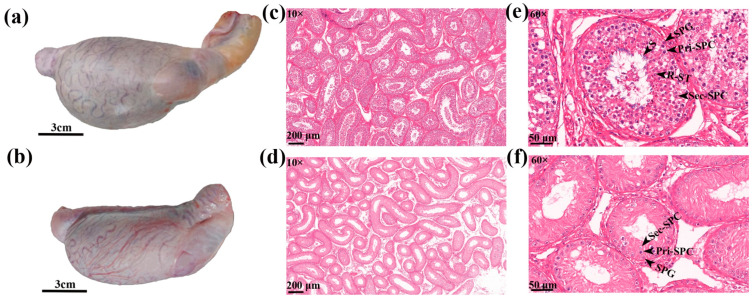
Histomorphological observation of testis. (**a**) Yak testis, (**b**) cattle-yak testis, (**c**) yak and (**d**) cattle-yak testis tissue sections 10×, and (**e**) yak and (**f**) cattle-yak testis tissue sections 60×.

**Figure 2 animals-12-02711-f002:**
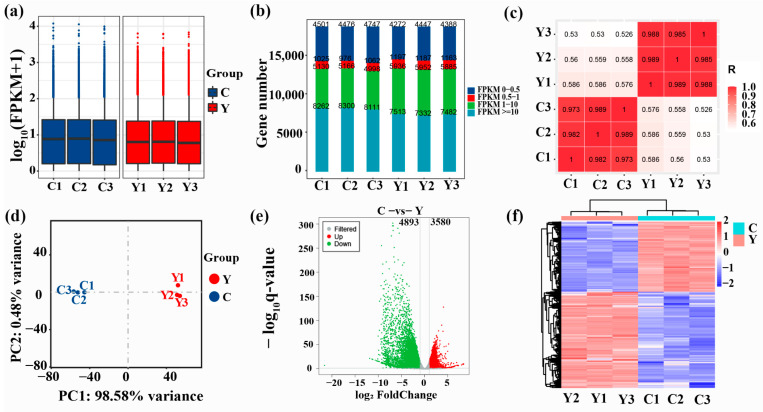
Protein-coding gene expression analysis. (**a**) Boxplot of gene FPKM values in respective samples, (**b**) FPKM expression distribution map, (**c**) heatmap showing correlation analysis, (**d**) PCA diagram, (**e**) differential expression volcano plot and (**f**) differential gene grouping cluster map (Protein-coding genes with comparatively high and low expression are denoted by red and blue, respectively). C: cattle-yak, Y: yak.

**Figure 3 animals-12-02711-f003:**
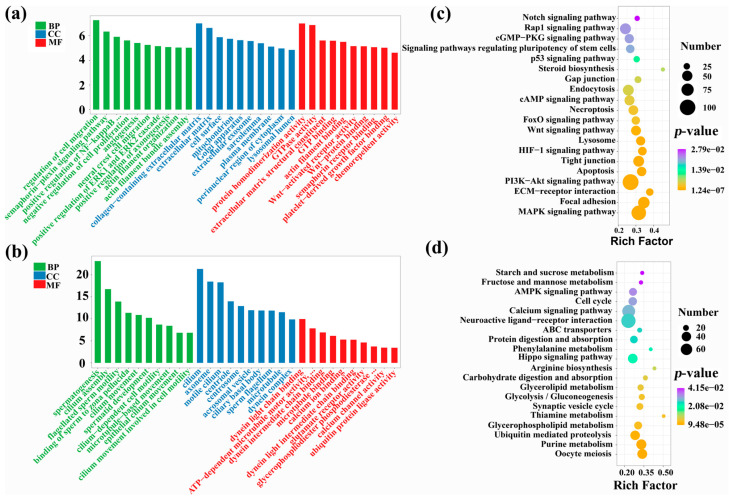
GO/KEGG analyses of DEGs. GO enrichment evaluations for (**a**) up-regulated and (**b**) down-regulated DEGs within cattle-yak. KEGG enrichment evaluations for (**c**) up-regulated and (**d**) down-regulated DEGs in cattle-yak. Go enrichment analysis ranks 10 entries from large to small according to the respective −log_10_p-value. KEGG analysis shows some significant enrichment results.

**Figure 4 animals-12-02711-f004:**
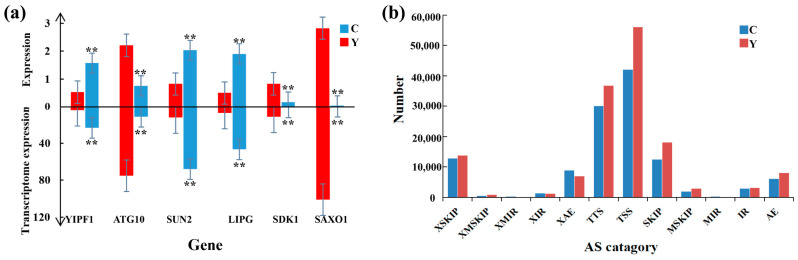
RT-qPCR validation of RNA-seq data and AS analysis. (**a**) RT-qPCR results for YIPF1, ATG10, SUN2, LIPG, SDK1, and SAXO1 genes within cattle-yak and yak groups. ** *p* < 0.01. (**b**) Level of differing AS events within cattle-yak/yak groups. C: cattle-yak, Y: yak.

**Table 1 animals-12-02711-t001:** Morphological parameters of testis.

Index	Yak	Cattle-Yak
Testicular circumference (cm)	14.75 ± 0.44 **	9.83 ± 0.51
Testicular length (cm)	12.31 ± 0.66 *	11.10 ± 0.75
Testicular weight (g)	88.77 ± 2.68 **	60.02 ± 2.20

Note: * *p* < 0.05, ** *p* < 0.01.

**Table 2 animals-12-02711-t002:** Histological parameters of testis.

Index	Yak	Cattle-Yak
Diameter of seminiferous tubules (μm)	238.47 ± 19.06 **	175.82 ± 18.41
Seminiferous tubule wall thickness (μm)	2.95 ± 0.55 *	2.41 ± 0.41
SPG diameter (μm)	13.26 ± 1.14 **	8.97 ± 0.76
Pri-SPC diameter (μm)	17.03 ± 0.98 **	14.04 ± 1.19

Note: * *p* < 0.05, ** *p* < 0.01.

**Table 3 animals-12-02711-t003:** Mapping results with yak reference genome.

Sample	Total Reads	Total Mapped Reads	Multiple Mapped Reads	Uniquely Mapped Reads
C1	46,495,404	43,312,320 (93.15%)	5,154,644 (11.09%)	38,157,676 (82.07%)
C2	46,067,720	42,690,813 (92.67%)	4,899,340 (10.64%)	37,791,473 (82.03%)
C3	45,021,512	42,007,249 (93.30%)	5,055,431 (11.23%)	36,951,818 (82.08%)
Y1	46,352,778	44,083,089 (95.10%)	5,540,645 (11.95%)	38,542,444 (83.15%)
Y2	46,675,930	44,335,303 (94.99%)	5,586,382 (11.97%)	38,748,921 (83.02%)
Y3	45,556,352	43,289,125 (95.02%)	5,512,059 (12.10%)	37,777,066 (82.92%)

## Data Availability

Raw reads of transcriptome sequencing of yak and cattle-yak testis are available at GEO (accession number: GSE208693). https://www.ncbi.nlm.nih.gov/search/all/?term=GSE208693%20.

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
