# Peer review of "F1 Male Sterility in Cattle-Yak Examined through Changes in Testis Tissue and Transcriptome Profiles"

_animals, 2022, doi:10.3390/ani12192711_

Round 1

Reviewer 1 Report

Review Manuscript Animals-1868662

“F1 male sterility in cattle-yak examined through changes in testis tissue and transcriptome profiles

General comments:

The aim of this paper was to analyze the differences of histology and gene expression in testicular tissue development between yak and cattle-yak by using histology and the RNA-seq of testicular tissue. The authors demonstrated that spermatogonia stem cell marker genes and apoptotic related genes are upregulated and differentiation and meiosis related genes are downregulated. However, spermatogenesis is thought to be accomplished by the interaction between somatic cells and spermatogenic cells, which determine the proliferation, differentiation and apoptosis. Therefore, the authors should provide the somatic cells (Sertoli cells, Leydig cells and myoid cells) data and discuss them with other authors reports. Furthermore, interpretation of the results should be more carefully and effectively. For publication of this manuscript, minor revisions are necessary.

Comments to the authors

Several parts of the manuscripts are still unclear. The reviewer recommends the author further revision for publish this work. Especially for the information about the RNA seq data of somatic cells should be added and discuss. Furthermore, Discussion is rather unclear, author should focus on the meaning of each molecules expression on testis and discuss about the way to rescue of infertility of cattle yak, summarize them and edit appropriately.

Specific comments

1.    The information about the season to collect each sample should be cited, because depending on the timing, maturation rate will be different in Yak and cattle-yak.

2.    The authors should include fixation methods. Fixative and time for immersion of tissue will affect the results. Furthermore, the thickness of the section should be inserted in the text.

3.    Shimazaki et al. (Reproduction in Domestic Animals, 2022) showed that hybrid cattle-yak testes maintained proliferation ability but less apoptotic ability in spermatogenic cells when compared to yaks of the same age. Authors should discuss about the apoptosis of spermatogenic cells in hybrid cattle-yak after consideration about the authors results came from all testicular cells not by only spermatogenic cells.

4.    Authors should discuss about the somatic cell roles for spermatogenesis to refer after showing the results.  For example, Leydig cells in yak- cattle (Sato et al., Reproduction in Domestic Animals, 2020). 

5.    To control the gene expression, epigenomic modulation is important. Please discuss it with proper references, for example, Phakdeedindan et al. (Reproduction in Domestic Animals, 2020) and Li et al. (Theriogenology, 2020)

Author Response

  1. The information about the season to collect each sample should be cited, because depending on the timing, maturation rate will be different in Yak and cattle-yak.

Response 1: Thank you for your comments. The sample collection season information has been supplemented in line 99 of the text. As follows: all samples were collected on June 10, 2021.

  1. The authors should include fixation methods. Fixative and time for immersion of tissue will affect the results. Furthermore, the thickness of the section should be inserted in the text.

Response 2: Thank you for your comments. Fixative and time for immersion of tissue, and slice thickness has been supplemented in line 115-118 of the text. As follows: The excised testicular tissue pieces were placed in the testis tissue fixative solution, and the fixative solution was changed at the 6th, 12th, and 24th hours, and then soaked in 75% alcohol. After dehydration with an automatic tissue dehydrator, after embed-ding in paraffin, the paraffin block was cut into 6 μm thick sections with a microtome.

  1. Shimazaki et al. (Reproduction in Domestic Animals, 2022) showed that hybrid cattle-yak testes maintained proliferation ability but less apoptotic ability in spermatogenic cells when compared to yaks of the same age. Authors should discuss about the apoptosis of spermatogenic cells in hybrid cattle-yak after consideration about the authors results came from all testicular cells not by only spermatogenic cells.

Response 3: Thank you for your comments. Apoptosis of spermatogenic cells is discussed in lines 349-353 of the text. As follows: Gfra1, CD9, SOHLH1, SALL4, ID4 and FOXO1 were marker genes of SSCs, all upregulated within cattle-yak testes. However, SPC marker genes Ccna1 and PIWIL1 were downregulated, and ST marker genes TNP1 and TXNDC2 were almost not expressed, which indicates that undifferentiated spermatogonia accumulate in the testis of the cattle-yak, whereas differentiated spermatogenic cells decrease.

  1. Authors should discuss about the somatic cell roles for spermatogenesis to refer after showing the results. For example, Leydig cells in yak- cattle (Sato et al., Reproduction in Domestic Animals, 2020).

Response 4: Thank you for your comments. Lines 242-250 in the text supplement the somatic RNA-seq data information, As follows :In somatic cells, sertoli cell (SC) marker genes BEX2 (brain expressed X-linked 2, WFDC2(WAP four-disulfide core domain 2), ITGA6 (integrin subunit alpha 6), ex-cept WFDC2, were significantly upregulated in cattle-yak testis tissue; Leydig cells (LC) marker genes such as IGF1 (insulin like growth factor 1), IGFBP3 (insulin like growth factor binding protein 3), IGFBP5, CYP11A1 (cytochrome P450 family 11 sub-family A member 1),CYP17A1, INHBA (inhibin subunit beta A), ARX (aristaless related home-obox) and TCF21 (transcription factor 21) were significantly upregulated except for IGFBP3; Myoid cell (MC) marker genes ACTA2 (actin alpha 2), MYH11 (myosin heavy chain 11), GLI1 (GLI family zinc finger 1), etc. were significantly upregulated in cattle-yak testis tissue. The role of somatic cells in spermatogenesis is discussed in lines 366-379. As follows: Testosterone (T) secreted by Leydig cells plays a key role in the development of the male reproductive system and the maintenance of normal spermatogenesis. We found that the expressions of CYP11A1 and CYP17A1 were significantly upregulated in the testis tissue of cattle-yak. Sato et al. found that the expression of cattle-yak 3β-hydroxysteroid dehydrogenase (3β-HSD) was increased and the expression of an-drogen receptor (AR) was decreased. T is cholesterol under the regulation of STAR (steroidogenic acute regulatory protein), its side chain is cleaved by CYP11A1 to form pregnenolone, and then 3β-HSD is converted into progesterone, which is finally catalyzed by CYP17A1 in the endoplasmic reticulum, and then combined with AR binding comes into play. Therefore, we speculate that cattle-yak LC secrete more T, but the T concentration in seminiferous tubules is low due to the decrease of AR expression. Narula et al. found that excessive secretion of T can prompt SC to secrete inhibin A, which selectively inhibits the synthesis and secretion of FSH in the anterior pituitary gland. So the imbalance of hormone regulation in seminiferous tubules is an important cause of spermatogenesis block.

  1. To control the gene expression, epigenomic modulation is important. Please discuss it with proper references, for example, Phakdeedindan et al. (Reproduction in Domestic Animals, 2020) and Li et al. (Theriogenology, 2020)

Response 5: Thank you for your comments. The regulation of genes by epigenetic modifications is discussed in lines 429-442. As follows: Although there are many studies on male sterility in cattle-yak, most of them are single omics or a combination of two omics. In order to systematically and comprehensively analyze the molecular mechanism of male sterility in cattle-yak, a systematic and comprehensive study of male sterility in cattle-yak should be carried out in com-bination with all omics, including epigenomics, proteomics, and metabolomics. Epi-genetic regulation not only regulates the development of the male reproductive system, but also controls the early stages of embryonic development and the process of gametogenesis, impaired epigenetic dysregulation disrupts human spermatogenesis, leading to male infertility and related spermatogenesis disorders. Phakdeedindan's re-search showed that during the maturation of yak, the regulation of AcK9 gene and the constant level of DNA methylation are both necessary conditions for spermatogenesis, and the inappropriate expression of AcK9 and DNA methylation may be the main factors affecting the normal development of cattle-yak germ cells factor. In the next step, we should integrate all the data on male sterility in cattle-yak and explore the molecular mechanism of male infertility in cattle-yak as much as possible.

Reviewer 2 Report

This is a well-written and very interesting article that studies the histomorphological and transcriptome differences in testis between sexually mature yak and cattle-yak. The introduction describes the context and importance of the study. Material and methods are clear and explain all experimental procedures. Results are clear and concise, and they are very well discussed and supported with interesting similar references.

I only suggest considering next minor grammar suggestions:

-       Line 116 = Replace “guideline” by “guidelines”.

-       Line 121 = Replace “are” by “were”.

-       Line 125 = Replace the period by a comma.

-       Line 125 = Replace “And” by “and”.

-       Line 314 = Replace “wu et al.” by “Wu et al.”.

-       Line 324 = Replace “MI et al.” by “Mi et al.”.

-       Line 404 = Replace “/” by “and”.

-       Lines 408-408 = I suggest to remove this sentence “In total, 8,473 DEGs (3,580 upregulated and 4,893 downregulated) were identified” because it’s a result.

-       Materials and methods = Please include all statistical procedures used in the study such as “Correlation analyses” and “PCA analyses”.

-       References = According to the guidelines described in the “Instructions for Authors”, Journal name should be abbreviated and written in italic style, and Journal year should be written in bold style.

Author Response

  1. Line 116 = Replace “guideline” by “guidelines”.

Response 1: Thank you for your advice. The article has been revised as requested.

  1. Line 121 = Replace “are” by “were”.

Response 2: Thank you for your advice. The article has been revised as requested.

  1. Line 125 = Replace the period by a comma.

Response 3: Thank you for your advice. The article has been revised as requested.

  1. Line 125 = Replace “And” by “and”.

Response 4: Thank you for your advice. The article has been revised as requested.

  1. Line 314 = Replace “wu et al.” by “Wu et al.”.

Response 5: Thank you for your advice. The article has been revised as requested.

  1. Line 324 = Replace “MI et al.” by “Mi et al.”.

Response 6: Thank you for your advice. The article has been revised as requested.

  1. Line 404 = Replace “/” by “and”.

Response 7: Thank you for your advice. The expressions "yak/cattle-yak" in the article have been replaced with "yak and cattle-yak".

  1. Lines 408-408 = I suggest to remove this sentence “In total, 8,473 DEGs (3,580 upregulated and 4,893 downregulated) were identified” because it’s a result.

Response 8: Thank you for your comments. The sentence " In total, 8,473 DEGs (3,580 upregulated and 4,893 downregulated) were identified” has been deleted as requested.

  1. Materials and methods = Please include all statistical procedures used in the study such as “Correlation analyses” and “PCA analyses”.

Response 9: Thank you for your comments. The relevant statistical procedures have been supplemented in lines 147-149 of the text. As follows: Principal components analysis(PCA)between two sets of samples was performed using the R package vegan. Using the R package stats to draw boxplots and correlation cluster heatmaps for each sample gene.

  1. References = According to the guidelines described in the “Instructions for Authors”, Journal name should be abbreviated and written in italic style, and Journal year should be written in bold style.

Response 9: Thank you for your comments. We have revised the references.
